# Long-read sequencing reveals complex patterns of wraparound transcription in polyomaviruses

Jason Nomburg[1,2,3], Wei Zou[4], Thomas C. Frost[1,3], Chandreyee Datta[5,6,7,8], Shobha Vasudevan[5,6,7,8], Gabriel J. Starrett[9], Michael J. Imperiale[4,10], Matthew Meyerson[1,2,11,12]*, James A. DeCaprio[1,3,12]*

1 Department of Medical Oncology, Dana-Farber Cancer Institute, Boston, Massachusetts, United States of America, 2 Broad Institute of MIT and Harvard, Cambridge, Massachusetts, United States of America, 3 Harvard Program in Virology, Harvard University Graduate School of Arts and Sciences, Boston, Massachusetts, United States of America, 4 Department of Microbiology and Immunology, University of Michigan, Ann Arbor, Michigan, United States of America, 5 Massachusetts General Hospital Cancer Center, Harvard Medical School, 185 Cambridge St, CPZN4202, Boston, Massachusetts, United States of America, 6 Department of Medicine, Massachusetts General Hospital and Harvard Medical School, Boston, Massachusetts, United States of America, 7 Center for Regenerative Medicine, Massachusetts General Hospital, Harvard Medical School, Boston, Massachusetts, United States of America, 8 Harvard Stem Cell Institute, Harvard University, Cambridge, Massachusetts, United States of America, 9 Laboratory of Cellular Oncology, CCR, NCI, NIH, Bethesda, Maryland, United States of America, 10 Rogel Cancer Center, Ann Arbor, Michigan, United States of America, 11 Department of Genetics, Harvard Medical School, Boston, Massachusetts, United States of America, 12 Department of Medicine, Brigham and Women's Hospital, Harvard Medical School, Boston, Massachusetts, United States of America

* matthew_meyerson@dfci.harvard.edu (MM); james_decaprio@dfci.harvard.edu (JAD)

**Data Availability Statement:** All raw RNA sequencing data are available in fastq format at the NCBI sequence read archive at BioProject PRJNA780012. The fast5 current data for all

## Abstract

Polyomaviruses (PyV) are ubiquitous pathogens that can cause devastating human diseases. Due to the small size of their genomes, PyV utilize complex patterns of RNA splicing to maximize their coding capacity. Despite the importance of PyV to human disease, their transcriptome architecture is poorly characterized. Here, we compare short- and long-read RNA sequencing data from eight human and non-human PyV. We provide a detailed transcriptome atlas for BK polyomavirus (BKPyV), an important human pathogen, and the prototype PyV, simian virus 40 (SV40). We identify pervasive wraparound transcription in PyV, wherein transcription runs through the polyA site and circles the genome multiple times. Comparative analyses identify novel, conserved transcripts that increase PyV coding capacity. One of these conserved transcripts encodes superT, a T antigen containing two RB-binding LxCxE motifs. We find that superT-encoding transcripts are abundant in PyV-associated human cancers. Together, we show that comparative transcriptomic approaches can greatly expand known transcript and coding capacity in one of the simplest and most well-studied viral families.

Nanopore experiments is available at BioProject PRJNA806471. All code used in this project can be found at the zenodo and github links below. The zenodo repository also contains all processed data necessary to reproduce all analyses and figures. The main processing steps used to process the RNAseq data are present as nextflow pipelines which call modular bash and python scripts. Zenodo: https://doi.org/10.5281/zenodo.5593468 Github: https://github.com/jnoms/SV40_transcriptome Furthermore, a series of interactive Google Colab notebooks can download all processed data from Zenodo and completely reproduce all analyses and non- schematic primary figures. The colab documents are stored on github at https://github.com/jnoms/SV40_transcriptome/tree/main/bin/colab. Direct links to the Google Colab documents are as follows: Fig 1: https://colab.research.google.com/github/jnoms/SV40_transcriptome/blob/main/bin/colab/Figure1.ipynb Fig 4: https://colab.research.google.com/github/jnoms/SV40_transcriptome/blob/main/bin/colab/Figure4.ipynb Fig 6: https://colab.research.google.com/github/jnoms/SV40_transcriptome/blob/main/bin/colab/Figure6.ipynb A Google Colab notebook is available for interactive investigation of all SV40 and BKPyV viral transcript classes, and does not require computational skills to use: https://colab.research.google.com/github/jnoms/SV40_transcriptome/blob/main/bin/colab/PyV_exploratory.ipynb

**Funding:** This work was supported in part by the US Public Health Service grants R35CA232128 and P01CA203655 and by the Bridge Project, a partnership between the Koch Institute for Integrative Cancer Research at MIT and the Dana-Farber/Harvard Cancer Center to J.A.D.; a Cancer Grand Challenges OPTIMISTICC team award (C10674/A27140) and by National Cancer Institute grant R35CA197568 to M.M.; R01 AI060584 and R21 AI147155 to M.J.I; CD & SV were supported by R35GM134944; GJS is supported by the NIH Intramural Research Program. The funders had no role in study design, data collection and analysis, decision to publish, or preparation of the manuscript.

**Competing interests:** I have read the journal's policy and the authors of this manuscript have the following competing interests: M.M. receives research support from Bayer, Janssen, Ono; consults for Bayer, Interline, Isabl; and receives patent royalties from Labcorp and Bayer. J.A.D. has received research support from Rain Therapeutics, Inc. and is a consultant for Rain Therapeutics, Inc. and Takeda, Inc.

## Author summary

Polyomaviruses (PyV) are small, double-stranded DNA viruses that cause devastating human diseases. Despite the clinical relevance of PyV, the full assortment of transcripts generated by PyV during infection is poorly characterized. We used long- and short-read RNA sequencing (RNAseq) approaches to greatly expand known transcript diversity of the human pathogen BK polyomavirus (BKPyV) and the prototype PyV simian virus 40 (SV40). Because PyV contain circular genomes, during transcription the host RNA polymerase can circle the genome multiple times in a process called wraparound transcription. We find that wraparound transcription is widely conserved across PyV and generates diverse early and late RNAs. We use short-read RNAseq from eight PyV to identify conserved but previously unannotated transcripts encoding novel gene products. One of these transcripts encodes superT, a T antigen that contains two RB-binding LxCxE motifs. We find that superT is expressed in PyV-associated human cancers. Together, this work expands our knowledge of PyV transcriptomes and identifies novel transcripts of potential relevance to human disease.

## Introduction

Polyomaviruses (PyV) are ubiquitous pathogens that can cause devastating human diseases [1] including polyomavirus-associated nephropathy (PVAN), hemorrhagic cystitis, and bladder cancer associated with BKPyV [2], progressive multifocal leukoencephalopathy caused by JCPyV, Merkel cell carcinoma caused by Merkel cell polyomavirus (MCPyV), and dermatosis caused by human polyomavirus 7 (HPyV7) [1,3]. PyV have circular double-stranded DNA genomes and express viral genes with distinct early and late kinetics. Early and late transcripts are driven by a bi-directional central promoter, and each terminate at their own polyA signal sequence located between the early and late regions [4,5]. The PyV early region encodes tumor or T antigens that promote cell cycle progression and facilitate replication of the viral genome by host DNA polymerase. The PyV late region originating from the common PyV promoter element on the opposite genome strand encodes the structural proteins required for the generation of progeny virions [4,5].

PyV transcripts undergo complex splicing to increase their coding capacity in the face of their small ~5kb genomes. In addition to the large T (LT) and small T (ST) antigens, additional T antigen splice forms have been identified including transcripts that generate truncated versions of LT (17kT, 57kT, truncT, and T' in SV40, MCPyV, BKPyV, and JCPyV respectively), a "superT" antigen that contains a duplicated LxCxE RB-binding motif in SV40, middle T (MT) in murine PyV (MPyV), and ALTO in MCPyV [6–13]. Although the diversity of late transcripts has been explored in SV40 [14], late transcript diversity in other PyV, including the major human pathogens, is poorly characterized. To address this lack of knowledge of PyV transcription and to discover unannotated biologically relevant PyV-encoded protein products, we used long- and short-read RNA sequencing technologies to characterize the transcriptomes of eight human and non-human PyVs.

## Results

### RNA sequencing expands PyV transcript diversity

To expand known PyV transcript diversity, we conducted a series of viral infections followed by total or polyA short-read Illumina RNA sequencing (short-RNAseq (total) and short-

RNAseq (polyA) respectively) (**Fig 1A**). We integrated this newly generated data with publicly available data from infected cell culture, human skin, and other settings (**Table 1**). Viruses studied include SV40, BKPyV Dunlop variant and Dik (wild type, or archetype), JCPyV, MPyV, MCPyV, HPyV7, and bark scorpion polyomavirus 1 (BSPyV1).

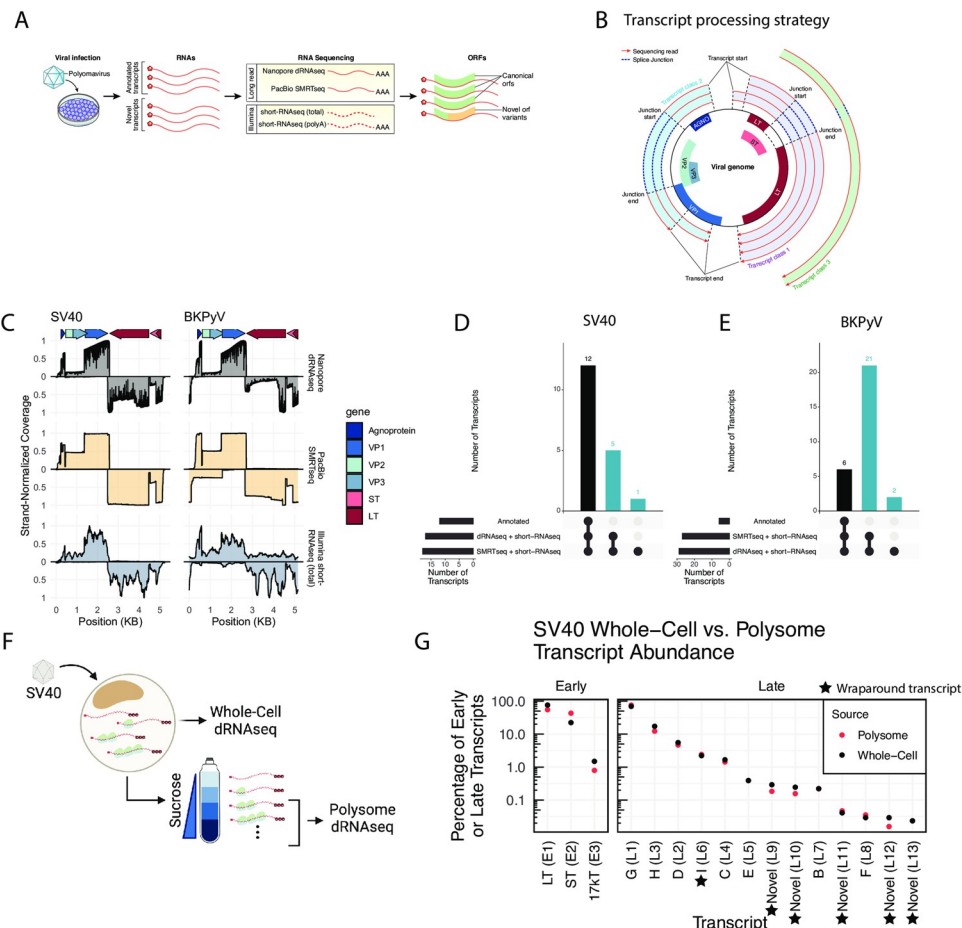

**Fig 1. RNA sequencing expands known SV40 and BKPyV transcript diversity. A.** Overview of experimental procedures. Cells were infected with a polyomavirus, and RNAs extracted. RNA was sequenced using long-read (Nanopore dRNAseq and PacBio SMRTseq) and short-read (Illumina short-RNAseq (total) and short-RNAseq (polyA)). Transcripts were analyzed, and the impact of observed splice events on viral open reading frames was assessed. **B.** Mechanism of transcript clustering in this study. Transcripts were aligned to the viral genome and grouped into transcript classes based on the presence of shared introns. Thus, within a transcript class there may be variation in the exact transcript start and end positions. This clustering strategy was used for both long- and short-RNAseq data. **C.** Viral RNA sequence coverage for SV40 and BKPyV as determined from dRNAseq, SMRTseq, and short-RNAseq (total) data. The Y axis indicates the scaled coverage, with X axis indicating the position on the viral genome. Coverage for late transcripts (mapping to the + strand) is above the x axis, while coverage for early transcripts (mapping to the − strand) is below the x axis. Coverage is scaled separately for each strand such that the maximum observed coverage for each strand is 1. Arrows at the top of the plot indicate the positions of viral genes. **D-E.** UpSet plot indicating the overlap between existing transcript annotations, dRNAseq data, and SMRTseq data for SV40 (**D**) and BKPyV Dunlop (**E**). Bars indicating overlap with existing transcript annotations are black, while those indicating no overlap with existing annotations are blue. These blue bars indicate the number of novel, unannotated transcripts identified. **F.** Overview of polysome profiling of SV40-infected cells. BSC40 cells were infected with SV40. Cells were lysed, and a portion of the lysate was subjected to dRNAseq (representative of the RNA content of the whole cell). The remaining lysates was centrifuged through a sucrose gradient, after which fractions containing RNA associated with two or more ribosomes were pooled and subjected to dRNAseq. Created with BioRender.com. **G.** Relative abundance of SV40 early and late transcripts in the whole-cell and polysome fractions of SV40-infected cells. Y-axis indicates the percentage of early or late transcripts and is log scale. X axis indicates each transcript, with black dots indicating each transcript's whole-cell relative abundance and red dots indicating each transcript's polysome relative abundance. A black star indicates the transcript is a wraparound transcript.

**Table 1. Information on all samples and viruses (excluding tumors) can be found in Table 1.**

| Virus | Sequencing Type | Origin (Accession) | MOI / Timepoint | Host |
|---|---|---|---|---|
| SV40 | dRNAseq (two replicates) | Generated here | MOI 1 / 48hpi | *C. Sabaeus* |
| SV40 | SMRTseq | Generated here | MOI 1 / 48hpi | *C. Sabaeus* |
| SV40 (polysome input/whole-cell) | dRNAseq | Generated here | MOI 1 / 44hpi | *C. Sabaeus* |
| SV40 (polysome) | dRNAseq | Generated here | MOI 1 / 44hpi | *C. Sabaeus* |
| SV40 | Short-RNAseq (total) | Generated here | MOI 1 / 48hpi | *C. Sabaeus* |
| SV40 | short-RNAseq (polyA) | Generated here | MOI 1 / 48hpi | *C. Sabaeus* |
| BKPyV (Dunlop) | dRNAseq | Generated here | MOI 0.5 / 3dpi | Human |
| BKPyV (Dunlop) | SMRTseq | Generated here | MOI 0.5 / 3dpi | Human |
| BKPyV (Dunlop) | Short-RNAseq (total) | Generated here | MOI 0.5 / 3dpi | Human |
| BKPyV (Dunlop) | short-RNAseq (polyA) | Generated here | MOI 0.5 / 3dpi | Human |
| BKPyV (Dik) WT | Short-RNAseq (total) | Generated here | MOI 1 / 5dpi | Human |
| BKPyV (Dik) WT | Short-RNAseq (polyA) | Generated here | MOI 1 / 5dpi | Human |
| BKPyV (Dik) M1 | Short-RNAseq (polyA) | Generated here | MOI 1 / 5dpi | Human |
| BKPyV (Dik) M2 | Short-RNAseq (polyA) | Generated here | MOI 1 / 5dpi | Human |
| MPyV | dRNAseq | Generated here | Unknown / 28hpi | Mouse |
| MPyV | Short-RNAseq (total) | Garren et al. [19] (SRR2043214) | MOI 50 / 36hpi | Mouse |
| JCPyV | Short-RNAseq (total) | Assetta et al. [40] (SRR9967610) | Unknown / 9dpi | Human |
| MCPyV (Synthetic genome) | short-RNAseq (polyA) | Theiss et al. [43] (EBI: ERS760222) | 200ng viral DNA / Unknown | Human |
| HPyV7 | Short-RNAseq (total) | Rosenstein et al. [44] (SRR11488976, SRR11488977) | From infected human skin | Human |
| BSPyV1 | Short-RNAseq (total) | Identified by Schmidlin et al. [45] (SRR5958578) | From whole scorpion | *C. sculpturatus* |

For SV40 and the BKPyV Dunlop variant, which replicate robustly in cell culture, we complemented short-read sequencing with Nanopore direct RNA sequencing (dRNAseq) and PacBio Single-Molecule Real-Time sequencing (SMRTseq) (**Fig 1A**), two long-read sequencing approaches for polyA RNA with distinct library preparations and sequencing strategies. Resultant RNAseq reads from long- and short-read sequencing strategies were mapped against the viral reference genome and grouped into transcript classes based on the presence of shared introns as detailed in the Methods (**Fig 1B**). For SV40, viral transcripts represented 11.6% and 8.8% of transcripts in dRNAseq and SMRTseq, respectively. For BKPyV Dunlop, viral transcripts represented 28.6% and 27.8% of transcripts in dRNAseq and SMRTseq, respectively. The total number of viral reads is detailed in **S1A Fig**. Transcripts within the same class contain the same introns but may have distinct transcript start sites (TSSs) and transcript end sites (TESs). For most transcriptomes, the majority of transcripts are members of the first few transcript classes (**S1B Fig**). To filter out erroneous splice sites, we required that all junctions detected in a dRNAseq or SMRTseq read must also be supported by at least 5 splice junction-spanning reads in short-RNAseq (total) data. Detailed information on this transcript class strategy is present in the Methods.

Comparison of read coverage from short-RNAseq (total), dRNAseq, and SMRTseq revealed that dRNAseq and SMRTseq were relatively consistent, with read coverage generally reflecting expected patterns of exon usage (**Fig 1C**). By contrast, the read coverage of short-RNAseq (total) was less representative of expected viral exon usage and may reflect noise due to the amplification of smaller RNA fragments (**Fig 1C**).

For SV40 and BKPyV Dunlop, a transcript class (consisting of transcripts with shared introns) was considered a bona-fide viral transcript if it comprised at least 0.1% of late or early transcripts in dRNAseq or SMRTseq data as described in the Methods. For SV40, which has detailed splice annotations [14], we found that dRNAseq and SMRTseq data are largely

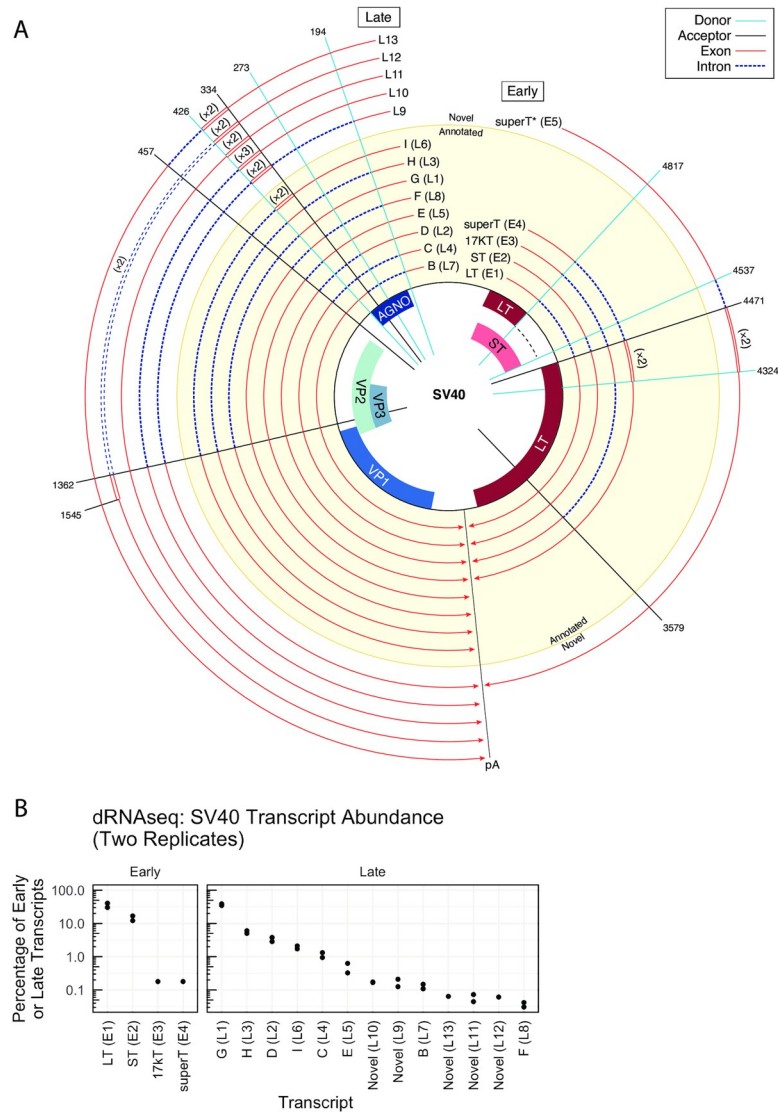

**Fig 2. Annotated and novel SV40 transcripts. A.** Transcripts are shown relative to the viral genome. Each line is a viral transcript, with red lines indicating exons and dashed blue lines indicating introns. Spokes indicate the positions of common splice donors and splice acceptors. Transcripts that were annotated prior to this study are on a yellow background, and novel transcripts are on a while background. Wraparound transcription that results in multiple copies of a region is annotated with double lines, and the number of copies is indicated in parentheses. The line labeled "pA" indicates the approximate position of the polyA signal sequence. **B.** The relative abundance of early or late transcripts in SV40 dRNAseq data. If a transcript was observed in SMRTseq but not dRNAseq, it is not present. The abundance of each transcript in both replicates of SV40 dRNAseq are plotted as individual dots.

consistent with existing annotations. However, we identified five previously unannotated SV40 transcripts that were supported by both long-read sequencing approaches, plus one additional previously unannotated SV40 transcript class supported by SMRTseq and short-RNA-seq (total) (**Figs 1D and 2).**

In contrast to SV40 and despite its clinical importance, BKPyV transcripts have been poorly characterized. We identified a total of 23 transcripts, 21 of which are supported by both dRNA-seq and SMRTseq data and only six of which were previously identified [8,15] (**Figs 1E and 3**). While novel BKPyV late transcripts are often analogous to the characterized wraparound and

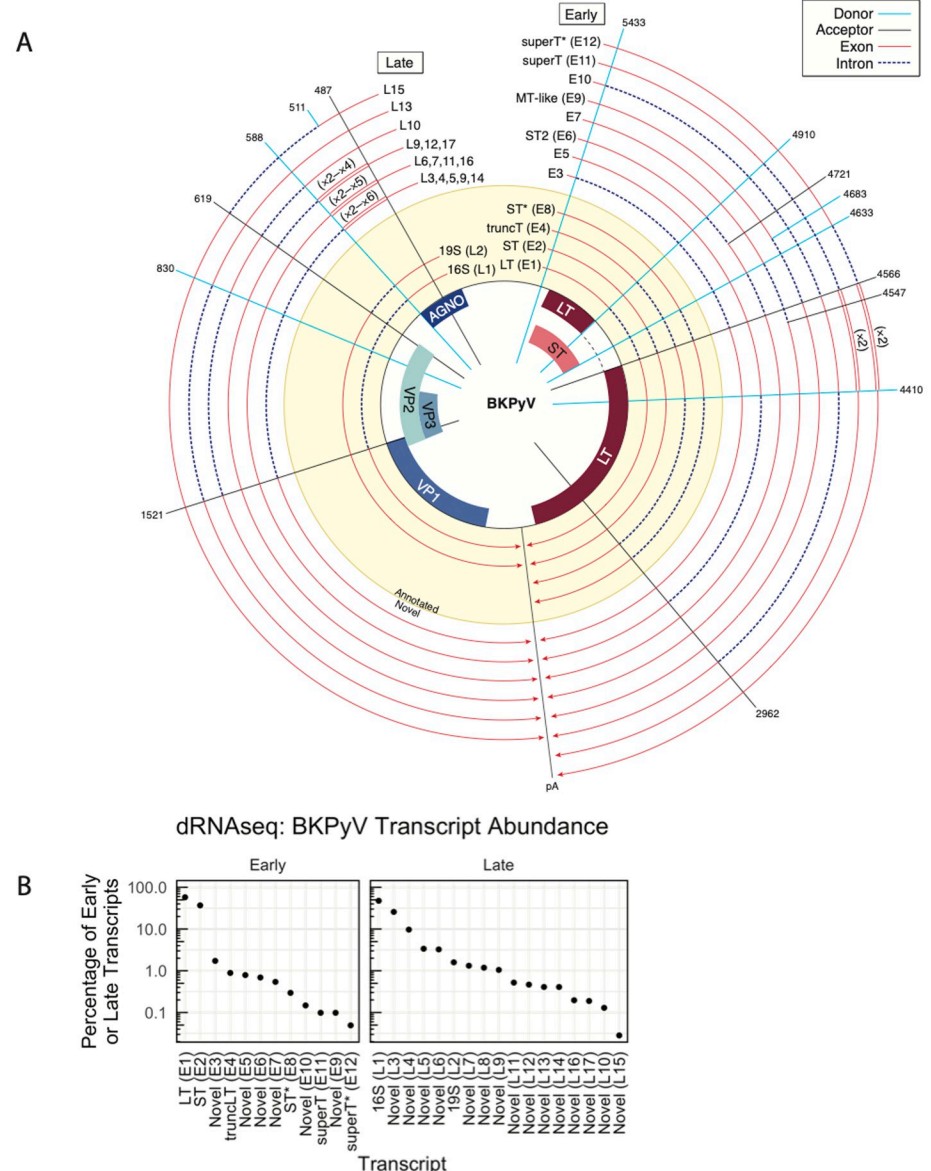

**Fig 3. Annotated and novel BKPyV transcripts. A.** Transcripts are shown relative to the viral genome. Each line is a viral transcript, with red lines indicating exons and dashed blue lines indicating introns. Spokes indicate the positions of common splice donors and splice acceptors. Transcripts that were annotated prior to this study are on a yellow background, and novel transcripts are on a while background. Wraparound transcription that results in multiple copies of a region is annotated with double lines, and the number of copies is indicated in parentheses. The line labeled "pA" indicates the approximate position of the polyA signal sequence. **B.** The relative abundance of early or late transcripts in BKPyV (Dunlop) dRNAseq data. If a transcript was observed in SMRTseq but not dRNAseq, it is not present.

non-wraparound transcripts previously identified in SV40, several additional and unexpected BKPyV early transcripts were identified. For example, an atypically early TSS revealed a splice donor that was used to generate transcript E3 (**Fig 3A**). Early transcripts including E6, E9, and

E11 are conserved across numerous PyV and lead to formation of novel ORFs—these are described in detail below.

We generated a comprehensive atlas of SV40 and BKPyV transcripts in **S3**–**S8 Figs**. Watch plots display the structure of each identified transcript, and read pileups show all transcripts identified in each transcript class. The relative abundance of each transcript as well as exact splice coordinates and abundance information for each identified transcript is provided in **S1 and S2 Tables**. Transcripts can also be explored using an interactive Google Colab notebook (https://colab.research.google.com/github/jnoms/SV40_transcriptome/blob/main/bin/colab/PyV_exploratory.ipynb). A comprehensive analysis of all splice sites detected in short-read short-RNAseq (total) and short-RNAseq (polyA) in eight PyV studied is presented in **S9 Fig**.

To address the possibility that distinct transcript isoforms could be preferentially translated, we performed polysome profiling of SV40-infected cells coupled with dRNAseq of whole-cell and polysome-associated polyadenylated RNAs (**Fig 1F**). The ribosome occupancy, determined as the ratio between a transcript's normalized polysome abundance and its normalized whole-cell abundance, has a mean of slightly above 1 for host transcripts (**S2D Fig**). We found 11.2% of reads in the whole-cell fraction and 18.7% in the polysome fraction were viral, consistent with active translation of viral transcripts. For late transcripts, the relative abundance in the whole-cell fraction was tightly coupled to polysome relative abundance (**Fig 1G**), indicating limited preferential translation of late transcripts. In contrast to late transcripts, we found that the LT:ST ratio was 1.3:1 in the polysome fraction compared to a 3.4:1 ratio of LT:ST transcripts in the whole-cell fraction, indicating preferential translation of ST during infection.

## Wraparound transcription is conserved across diverse PyV

Long-read sequencing revealed the existence of many late transcripts that contain multiple copies of a duplicated leader exon. Leader-leader splicing is due to "wraparound transcription" of PyV transcripts that failed to terminate at the late polyadenylation signal and continue to circle the genome repeatedly. RNAs containing repetitive leader sequences were originally observed in MPyV by Acheson in 1978 [16] and subsequently described by others, although the structure and diversity of these RNA species is unknown [17–20]. We investigated these transcripts in dRNAseq data from SV40 and BKPyV. To supplement these data, we also performed dRNAseq on MPyV-infected cells. Wraparound transcription, defined by the presence of repetitive copies of a shared leader sequence, was found in long-read sequencing for all three PyVs (**Fig 4A, 4B and 4C**: note the presence of the leader-leader or repeated exon near the "11 o'clock" position in watch plots). In addition to this leader sequence repetition, there were diverse forms of wraparound transcripts that contain various combinations of subsequent introns and encode for distinct viral proteins (**Figs 2 and 3**). While only 3.6% of SV40 transcripts originate from wraparound transcription, BKPyV and MPyV have markedly higher rates at 25% and 41% respectively (**Fig 4D**). Notably, we find that there is limited difference in the polysome and whole-cell relative abundance of wraparound transcripts (**Fig 1G**), suggesting that leader-leader splicing has little effect on translation.

Next, we inferred the presence of wraparound transcription in diverse PyV by identifying short-RNAseq (total) reads that span the leader-leader junction (**Fig 4E**). Despite the limited length of these short reads, leader-leader junctions can be accurately identified within a single read through analysis of junction sites (**Fig 4F**). We found evidence of wraparound transcription in all eight PyV investigated here. This includes HPyV7 RNAseq from infected human skin and RNAseq data from a scorpion containing the highly divergent Bark scorpion polyomavirus 1 (BSPyV1), indicating that wraparound transcription occurs *in vivo* and is widely conserved across PyV.

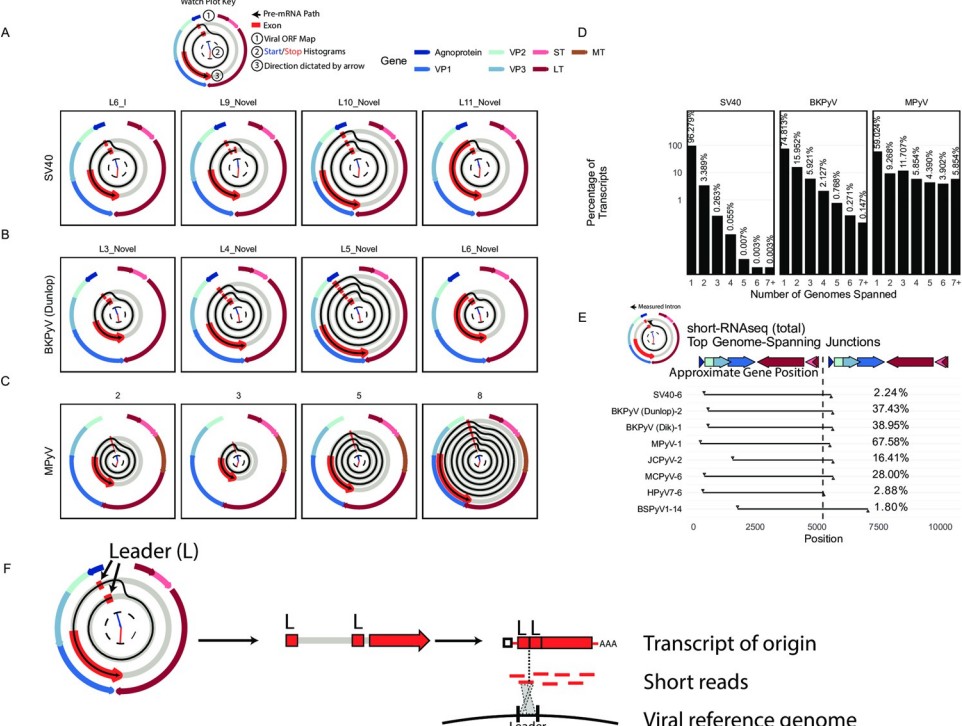

**Fig 4. Pervasive wraparound transcription across PyV. A-C.** Watch plots indicating the top 4 highest abundance late wraparound transcript classes in dRNAseq data from SV40 **(A)**, BKPyV Dunlop **(B)**, and MPyV **(C)**. The outer ring of each watch plot indicates the position of the viral ORFs. The inner arms are histograms detailing the distribution of transcript starts (in blue) and ends (in red) for transcripts within each transcript class. The red segments indicate exons. Transcripts start in the innermost ring—a second or third ring indicates that the pre-mRNA that generated the transcript must have circled the viral genome multiple times. The 3' end of the transcript and the direction in which these plots are oriented is indicated by the red arrow at the end of the last exon segment. The red exon segments start at the most common transcript start site within the transcript class, and end at the most common transcript end site within the class. The watch plot key shows an example of the path of the pre-mRNA for SV40 transcript class L6_I. **D.** Bar plots indicating the percentage of late transcripts that span a given number of genome lengths in SV40, BKPyV Dunlop, and MPyV dRNAseq data. **E.** The leader-leader junction, that connects the pre-mRNA from one genome to the subsequent wraparound, was identified in Illumina short-RNAseq (total) data. The intron in question is plotted as a black line in this plot, with the x axis indicating the genomic position of the intron. The top late wraparound transcript for each virus was plotted. The gene map indicates the approximate gene position and is accurate for SV40— the exact position of the viral genes varies between viruses. Percentages indicate the percentage of late junction-spanning transcripts that support the plotted wraparound leader-leader junction. **F.** Schematic illustrating how leader-leader wraparound transcription can be detected from short read short-RNAseq (total). Leader-leader splicing can be seen as a repetitive exon in watch plots from long-read RNAseq data. Ultimately, there was an original processed mRNA in the cell that contained two tandem leader sequences. When this transcript of origin is sequenced via short read sequencing, reads will be generated across its length. A minority of these reads will span the leader-leader junction, and mapping against the viral reference genome can be used to uncover leader-leader splicing.

## Pervasive premature polyadenylation of early transcripts in SV40, BKPyV, and MPyV

We found that many early transcripts in SV40 and BKPyV underwent alternative polyadenylation (APA) earlier than the canonical polyA site as indicated by premature transcript end positions near 3 o'clock in the watch plots (**S10A and S10B Fig**). Early transcript APA had been previously identified in MPyV, where there is a canonical polyA signal sequence (AATAAA) within the LT ORF [21,22]. Indeed, dRNAseq identified APA of early transcripts in MPyV-infected cells (**S10C and S10D Fig**). In contrast to MPyV, APA in SV40 and BKPyV may be

driven by alternative polyA signal sequences to the 5' of the APA site (ATTAAA in SV40, AAGAAA or TATAAA in BKPyV). Assessment of the cumulative incidence of early transcript termination shows abrupt increases in transcript termination ~1500nt upstream of the canonical polyA site in all three viruses (**S10D Fig**). This APA appears to be similarly abundant in LT and ST transcripts. We found that transcripts with APA still contain a full polyA tail that, while shorter than the polyA tails of transcripts that use the canonical polyA site, still tend to be longer than the polyA tails of host transcripts (**S10E, S10F, S10G, and S2C** Figs). The polyA tail length of a spike-in control RNA with a known 30-adenine polyA tail was correctly estimated by dRNAseq (**S2C Fig**). We find that transcripts containing APA can associate with polysomes (**S10H and S10I Fig**), indicating that these transcripts are translated.

## Comparative analysis of short-RNAseq (total) data reveals conserved, unannotated splice-forms that may generate variant ORFs

Next, we conducted a comparative analysis of PyV transcription from short-RNAseq (total) data (**S9 Fig**), with the hypothesis that data from diverse PyV could reveal unannotated splice forms. This analysis led to the discovery of several unannotated but conserved splicing events that have the potential to expand the coding capacity of PyV (**Fig 5**).

We found that PyVs including HPyV7, MPyV, BKPyV Dunlop, and MCPyV express a transcript utilizing the LT first exon donor but an acceptor within the ST ORF leading to the generation of the ST2 ORF (**Fig 5A**). This splice occurs in-frame in HPyV7 and BKPyV resulting in an internal deletion within ST, while in MPyV and MCPyV this splice lands out of frame and results in the addition of novel C-terminal amino acids. The ST2 splice is highly abundant in HPyV7 representing over 20% of spliced early transcripts from HPyV7-infected human skin. ST2-encoding transcripts were detected in BKPyV dRNAseq and SMRTseq data (transcript E6).

MPyV encodes MT in addition to the LT and ST antigens common with other PyV. MPyV MT is generated from a splicing event that connects the ST ORF with an ORF in the alternative frame of the LT second exon. To our surprise, we found that BKPyV expresses low levels of a similar transcript containing a splice that connects the ST ORF with an MT-like ORF likewise in an alternative frame of the LT second exon (**Fig 5B**). This MT transcript was also detected in BKPyV dRNAseq and SMRTseq data (transcript E9).

JCPyV encodes two VP1 variants, VP1Xs, that consist of the N-terminal region of VP1 with novel C-termini that make up as much as 30% of late spliced transcripts in JCPyV (**Fig 5C**) and have been recently identified and validated by an independent group [23]. We found that VP1X-encoding transcripts were also produced by MCPyV, SV40, BKPyV, and MPyV, albeit at a lower abundance than in JCPyV. Except for one JCPyV VP1X-encoding splice, these transcripts were generated from splicing of wraparound transcripts that run through the late polyA signal sequence.

## SuperT, a T antigen containing two RB-binding motifs, is present in multiple PyV and in PyV-associated human cancers

Studies in SV40-transformed cells previously identified a superT antigen with higher molecular weight than LT, containing a duplicated region with two copies of the LxCxE RB-binding motif [24]. We found that a superT-specific splice was present in SV40, BKPyV (Dik and Dunlop variants), JCPyV, and MCPyV during viral infection (**Fig 5D**). The superT-specific splice originates from a splice donor canonically associated with a conserved truncated LT antigen (17kT in SV40, truncT in BKPyV, 57kT in MCPyV, and T' in JCPyV), but uses the LT second exon acceptor available due to wraparound transcription. We find evidence of superT in the

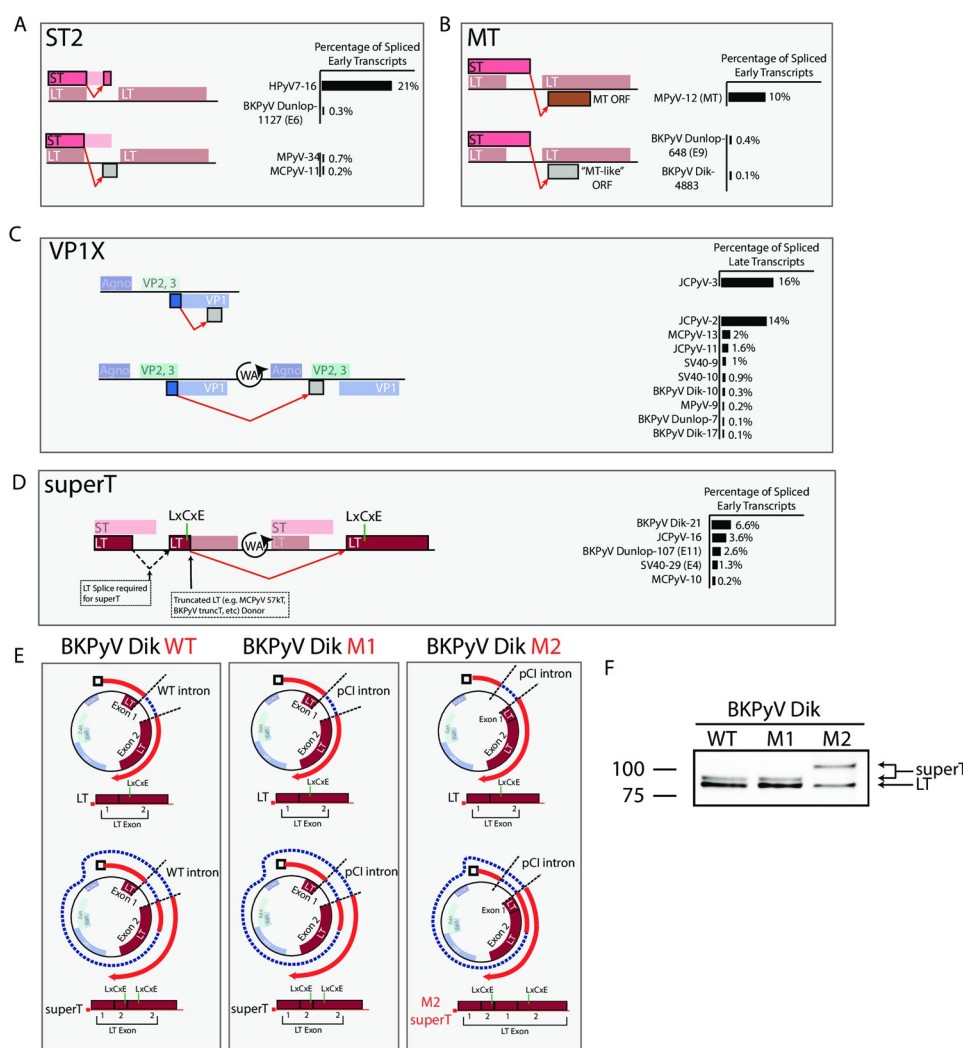

**Fig 5. Detection of novel, conserved splicing events that expand PyV coding capacity. A-D**. Schematics illustrating identified ORFs. Each row is a reading frame (except for ST and the LT 1st exon, which are in the same frame), and unannotated amino acids are represented by grey boxes. The measured intron is indicated by the red arrow. Colored ORFs are annotated, while grey ORFs are unannotated. Percentages on the right side of the Fig are the percentage of spliced viral transcripts on the same strand as determined from short-read short-RNAseq (total) data. Numbers after each virus name indicate the transcript class within each short-RNAseq (total) dataset. The measured intron is indicated by the red arrow. **A)** ST2: This ORF is generating from a splicing event that uses the LT first exon donor and an acceptor within the ST ORF. In HPyV7 and BKPyV Dunlop, the splice lands in frame and results in an internal deletion within ST. In MPyV and MCPyV the splice lands out of frame, resulting in an ORF that contains the N-terminal region of ST and novel amino acids at the C terminus. **B)** MT: MPyV encodes a MT following splicing connecting the end of the ST ORF with an ORF in an alternate frame of the LT second exon. In BKPyV, a similar splice occurs connecting ST with an MT-like ORF in an alternative frame of the LT second exon. **C)** VP1X: JCPyV encodes two VP1X ORFs generated by splicing within VP1 and landing in an alternative frame of VP1, or earlier in the late region due to wraparound transcription. While predominant in JCPyV, VP1X is likewise present in many other PyV. **D)** superT: The superT-specific splice utilizes the splice donor canonically associated with truncated T antigens such as 17kT in SV40 and truncT in BKPyV. Due to wraparound transcription, a LT second exon acceptor is available to the 3' of this donor and acts as the acceptor. For the superT ORF to form, an initial LT splice is required. Ultimately, superT contains a duplication in part of the LT second exon that includes the RB-binding LxCxE motif. **E.** Schematics detailing BKPyV Dik isolates used for querying the existence of superT. BKPyV WT is wild type virus. M1 contains a LT intron that has been replaced with an intron from the plasmid pCI. Both WT and M1 are expected to generate LT and superT of expected sizes. M2 has a completely removed LT intron, and the pCI intron is located directly 5' of the LT ORF. M2 is expected to encode LT of expected size, but a larger superT variant due to incorporation of a second copy of the LT first exon. **F.** Western blot of cells infected with BKPyV Dik WT, M1, or M2 and probed with an antibody reactive against LT. The lower molecular weight band is LT, and the higher molecular weight bands are consistent with superT.

dRNAseq and SMRTseq data for SV40 and BKPyV Dunlop infections (transcripts E4 and E11 respectively). Western blot with an antibody reactive to LT in BKPyV Dik-infected cells revealed a band with slightly higher molecular weight than LT that is consistent with superT (**Fig 5F**). BKPyV Dik mutant M1, designed to remove ST by replacing the LT intron with an intron from the plasmid pCI (**Fig 5E**), also generated a superT band of expected size. BKPyV Dik mutant M2 was generated by removing the LT intron and adding the pCI intron just 5' of the LT first exon. Should the truncT donor be used to generate superT in this mutant, the only available acceptor is before the LT 1$^{st}$ exon, which would result in the formation of an aberrantly larger superT due to the inclusion of a second copy of the LT first exon (**Fig 5E**). Short-RNAseq (polyA) analysis of cells infected with BKPyV Dik WT, M1, or M2 show junctions consistent with this model (**S11 Fig**), and western blot revealed that the superT band in M2 is shifted to a higher molecular weight (**Fig 5F**). Together, these data indicate that superT is generated by BKPyV Dik during viral infection.

SuperT was initially identified as an unexplained higher-molecular weight T antigen present in many SV40-transformed cell lines [12,13]. While superT can be generated during viral infection because of wraparound transcription, in SV40-transformed cells it would be possible to yield pre-mRNAs that can be spliced to form superT should the virus be integrated in tandem copies (**Fig 6A**). Indeed, we previously observed that MCPyV integration events in Merkel cell carcinoma (MCC) often lead to partial duplications of the viral genome and result in the tandem insertion of multiple copies of viral early genes [25]. Furthermore, the duplicated region in superT includes the RB-binding LxCxE motif, raising the possibility that superT can function as a potent oncogene. We therefore asked if there is evidence of superT in PyV-associated human cancers.

To address this question, we first analyzed short-RNAseq (total) data from five BKPyV-associated bladder cancers [2]. To our surprise, we found that short-RNAseq (total) data from two replicates of one BKPyV-associated bladder cancer contained a higher abundance of superT-specific splice compared to the LT- or ST-specific splices, suggesting that a large fraction of "LT" in this tumor is superT (**Fig 6B**). We next analyzed short-RNAseq (polyA) data from a series of 30 MCPyV-positive MCCs and found evidence of superT in six samples (**Fig 6B**). Notably, the total number of viral reads in some MCPyV-positive but superT-null tumors was very low, leaving open the possibility that sequencing depth was insufficient to identify the superT splice in additional tumors. Using PCR and sanger sequencing, we confirmed the presence of the superT splice in MCC tumor J45_440 (**S12A Fig**).

We hypothesized that superT may be generated by cis-splicing due to concatemeric integration of multiple copies of the etiologic PyV in these tumors (**Fig 6A**). To address this hypothesis, we investigated three MCCs (J45_440, J17_296, J11_285) for which we possess short-read whole genome sequencing data. From J11_285, we were able to assemble the entire integration site, showing that MCPyV is integrated in a manner that could allow cis-splicing to generate superT (**S12B Fig**). For J45_440, we assembled a single viral block integrated in chromosome 7 (**S12C Fig**). We found that 1) there are likely 2 copies of the viral genome, and 2) the 5' viral integration site appears to fall on chromosome 7 "after" the 3' viral integration site, observations consistent with the existence of two copies of the viral genome in tandem separated by a small segment of host DNA at this integration site. For J17_296, from the assembly, we could infer three distinct segments of viral DNA with integration sites closely spaced within chromosome 2 (**S12D Fig**), indicating a complex integration pattern. The longer block contains two copies of the early region and can likely support superT generation through cis-splicing. The LT ORF of MCPyV is often truncated by premature stop codons or deletions in MCC. We found that a stop codon in J17_296 likely prevents expression of superT, but no stop codons

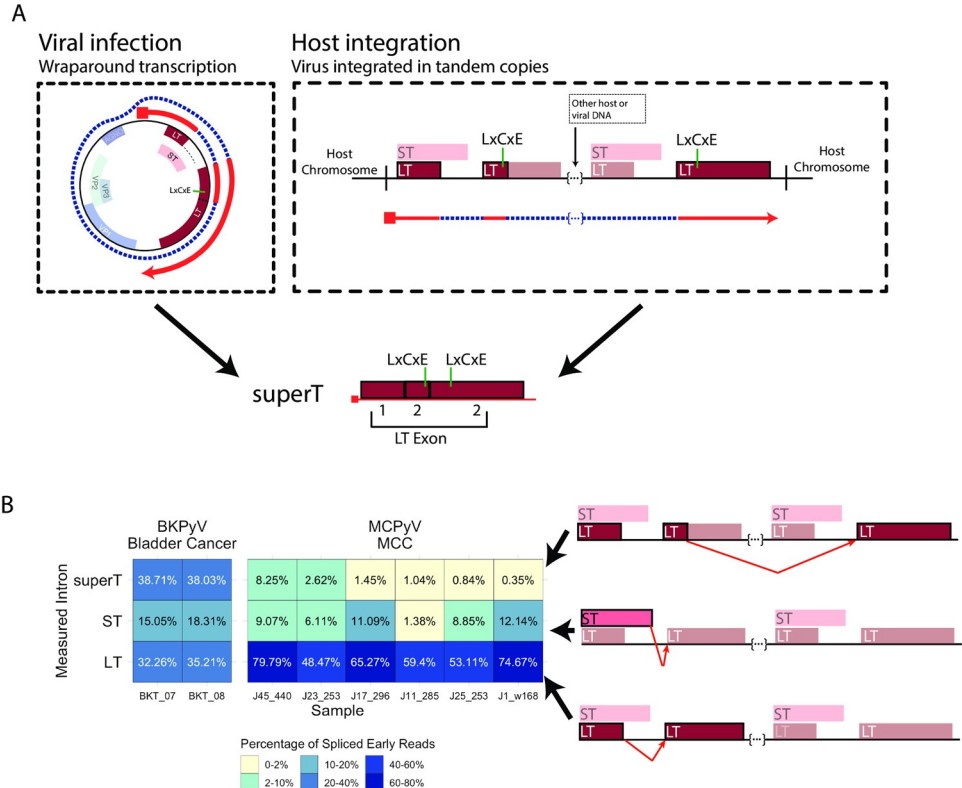

**Fig 6. Detection superT-encoding transcripts in PyV-associated cancers. A.** Schematic detailing the generation of superT during lytic infection and from integrated virus in cancer. During viral infection, the RNA polymerase can circle the viral genome multiple times, resulting in a pre-mRNA that can be spliced to generate superT. In the case of host integration, a polyomavirus can be integrated in tandem copies such that a pre-mRNA is generated with more than one copy of the viral early region. This pre-mRNA can be similarly spliced to generate a superT transcript. **B.** Heatmap indicating the abundance of the superT, ST, and LT introns from RNAseq data from two replicates of a BKPyV-positive bladder cancer and six MCPyV-associated MCCs. Percentages indicate the percentage of spliced early viral reads for each sample. The splice measured in each row is indicated by the red arrow in the schematics on the right side of the Fig.

occur before the superT splice in J45_440 or J11_285 (**S12E Fig**). Together, these data indicate that viral integration sites often could support cis-splicing to generate superT.

Two recent studies have found evidence of circular RNAs (circRNAs) that may be generated by MCPyV in MCC and may support the translation of ALTO [26,27]. Of note, the major circRNA splice is equivalent to our proposed MCPyV superT splice—a short-RNAseq read spanning the proposed circRNA junction cannot be differentiated from a read spanning the superT junction. However, the MCC RNAseq samples in which we found superT are short-RNAseq (polyA), which should select against the potential circRNA due to its lack of a polyA tail. Furthermore, we detect the superT splice in short-RNAseq (polyA) of SV40 and BKPyV Dunlop infections in cell culture (**S9 Fig**), although at around ~2/3 of its relative abundance in short-RNAseq (total). Finally, we identify full-length superT transcripts in dRNAseq data, which is highly unlikely to sequence circRNA since it is not polyadenylated. This leaves open the possibility that some superT-like splice in short-RNAseq (total) from viral infection originates from circRNA but suggests that most are from linear transcripts that contain a polyA tail.

## Discussion

Here, we show that leveraging multiple long- and short-read RNA sequencing approaches across 8 polyomaviruses has allowed us to greatly expand known transcript diversity of this viral family. Short read RNAseq has limited capacity to characterize transcriptome diversity because only a small fraction of reads span splice junctions, and these junctions often cannot be phased with other junctions or to the transcript start and end sites. Integrating long-read sequencing has allowed sequencing of entire transcripts, including phasing of splice sites and transcript start and end positions. Recent studies have leveraged long read sequencing to shed light on exceptional complexity in the transcriptomes of diverse RNA and DNA viruses [28–34]. We have expanded these studies to show that a comparative approach within a viral family can identify conserved transcripts that extend viral coding capacity.

Historically, studies of PyV transcripts were limited by the sensitivity and resolution of northern blots, or by the read length of short read sequencing. Despite these limitations, studies in the 70's and 80's were able to cumulatively characterize several SV40 late transcripts, including one containing leader-leader splicing [14,18,35]. In contrast to SV40, the architecture of BKPyV late transcripts is poorly characterized—prior to this work, the two major classes of late transcripts ("16S" and "19S", reflecting transcript size based on gradient sedimentation properties) were the primary late transcript classifications [15]. Only recently did a study provide some evidence for leader-leader splicing in BKPyV [36]. While the late transcripts of most PyV are thought to encode the canonical late viral proteins, a recent study in JCPyV identified two splice events that lead to the generation of novel proteins containing the N-terminal region of VP1—one of which was validated through western blot [23]. We found that these transcripts (deemed "VP1X") are highly expressed in JCPyV but are also expressed at lower level in BKPyV, MPyV, MCPyV, and SV40.

Leader-leader splicing is known to be highly prevalent in MPyV, where as many as 12 leader exons have been observed on a single RNA [37–39]—in our data, we have identified over 15 leader exons in a single transcript. Furthermore, leader-leader splicing is required for stable accumulation of MPyV late transcripts, dependent on length but not nucleotide composition of the leader [17]. Despite these observations, the exact structure, diversity, and conservation of wraparound transcripts was not understood. Here, we found that leader-leader splicing and wraparound transcription occurs in all PyV studied, including in the divergent Bark scorpion polyomavirus 1, and found that the prevalence of leader-leader splicing varies significantly between PyV. It is possible that this variation reflects differences in the strength of the late polyA signals of these PyV. We found a large diversity of wraparound transcripts containing variable numbers of the leader sequence and diverse patterns of subsequent exon usage. Finally, analysis of publicly-available temporal RNAseq data from JCPyV, BKPyV, and MPyV infection [19,40] suggests limited changes in wraparound abundance over time.

While late and early transcripts are thought to primarily end at the canonical late or early polyadenylation sites, studies previously observed APA of early transcripts in MPyV [21,22]. Here, we likewise identify pervasive APA of early SV40 and BKPyV Dunlop. We observe less early transcript APA in MPyV despite presence of a canonical AATAAA polyadenylation motif. Notably, the MPyV dRNAseq data is derived from cells at a relatively early stage of infection, raising the possibility for temporal regulation of early transcript APA. Future studies are required to investigate the factors influencing APA incidence in polyomaviruses.

We find that SV40 early transcripts with APA can associate with polysomes and are likely translated. In addition, polysome profiling revealed that SV40 transcripts are higher abundance in polysome-associated RNAs than in whole-cell RNA populations, indicating preferential translation of SV40 transcripts. The relative abundance of individual late viral transcripts

in the polysome closely reflected their whole-cell abundance—conversely, ST transcripts were preferentially translated compared to LT transcripts. The mechanism driving this difference needs further study, as these transcripts differ only by a minor difference in splice donor usage.

In addition to the major early transcripts encoding LT and ST, other early transcripts have been identified in some PyV. MPyV encodes MT, generated by a splice connecting the ST ORF and an ORF overprinted with LT second exon. MT is a primary oncogene in MPyV and was thought to be largely restricted to rodent PyVs [41]. We found that BKPyV generates a MT-like ORF through splicing connecting ST and an ORF similarly overprinted with the LT second exon, showing that non-rodent PyVs may be capable of expressing MT-like ORFs. In addition, MPyV also encodes a tinyT antigen consisting largely of the LT first exon, resulting from a splice connecting the LT first exon donor and MT acceptor [42]. We identified a novel T antigen, ST2, that is generated from a splice from the LT first exon donor to a splice acceptor within the ST ORF. This transcript is highly expressed in HPyV7 and present at lower levels in BKPyV, MPyV, and MCPyV. Many PyV encode a truncated variant of LT—this includes SV40 17kT, BKPyV truncT, MCPyV 57kT and JCPyV T' proteins [6–9]. These transcripts contain a canonical LT splice and a subsequent splice that removes a large portion of the LT ORF.

We found that the same secondary splice sites responsible for truncated LT variants can be used to generate superT. SuperT was initially observed in many SV40-transformed cell lines [12,13]—in a similar manner, we find that concatemeric integration of BKPyV and MCPyV in human cancers can facilitate the generation of superT. We also find that superT is generated in lytic infections of SV40, BKPyV, MCPyV and JCPyV. Eul and colleagues have published several studies proposing that SV40 superT can be generated by trans-splicing between two separate pre-mRNAs in the context of artificial expression constructs encoding the SV40 early region (20, 31, 32). However, we find that in MCC tumors that generate superT and for which we can assemble the viral integration site, the viral genome is likely integrated in tandem in a way that could facilitate the cis-splicing of pre-mRNA that spans multiple genome copies. Thus, while we cannot rule out trans-splicing from these data, we believe cis-splicing is more likely. Future studies are necessary to understand the biology of superT including its oncogenic potential and ability to bind multiple RB molecules. Finally, efforts should be taken to understand if superT is expressed by PyV and contributes to disease in other contexts, such as by BKPyV in PVAN or JCPyV in PML.

We show that complex, uncharacterized splicing events are used by PyV to expand their protein coding capacity. Future work is necessary to understand the biological function of these transcripts and proteins. It is possible that unannotated splicing we identify here could be differentially abundant in other biological contexts, so it will be important to investigate PyV splicing in other infection contexts and human diseases. Future transcriptome analyses that integrate long and short reads from multiple viruses may have utility to expand characterized transcript and coding capacity in other viral families.

## Conclusions

We provide a comprehensive transcriptome atlas for the prototype PyV SV40, as well as the critically important human pathogen BKPyV. Comparative analyses of PyV transcriptomes reveals conserved splice events that may expand PyV coding capacity. We find that superT, a transcript generated by SV40, BKPyV, JCPyV, and MCPyV that encodes a T antigen containing two RB-binding LxCxE domains, is present in several PyV-associated human cancers. Together, these data expand our understanding of PyV transcriptomes and uncover unannotated PyV-encoded proteins of potential relevance to human disease.

## Methods

### Datasets

**Tumor samples.** The BKPyV-associated bladder cancer is sample TBC03 that has been described [2]. This sample is stranded, short-RNAseq (total).

Merkel cell carcinoma samples: Sections of tissue were isolated from patient-derived tumor biopsies and suspended in RNAlater (Thermo Fischer) until further processing. RNA and DNA was extracted from each section via the AllPrep DNA/RNA kit (Qiagen). Isolated RNA and DNA were each sequenced (PE150) on the NovaSeq 6000 platform (Illumina) for a depth of 50 M reads or 60x genomic coverage per sample, respectively (Novogene). RNAseq data are unstranded, short-RNAseq (polyA). Note that samples J23_253 and J25_253 are from the same patient but were collected at different times.

**SV40 infection and RNA extraction.** BSC40 cells (ATCC CRL-2761) were seeded on 150mm dishes at $5.37*10^6$ cells per plate—about 70% confluence. After waiting 4 hours for the cells to adhere, cells were infected with SV40 at MOI 1 as previously described [46] with slight modification. In brief, maintenance media was removed, and each 150mm dish was inoculated with 6mL of virus stock diluted in DMEM + 2% FBS. Infection was allowed to proceed at 37˚C, 5% CO2 for one hour, with the plates rocked every 15 minutes to ensure adequate coverage of the solution over the cell monolayer. At the end of this period, DMEM + 2% FBS was added to a final volume of 25mL per 150mm dish. Each dish was then incubated at 37˚C, 5% CO2 for 48 hours. RNA was extracted using the QIAGEN RNeasy Mini Plus Kit (QIAGEN 74134). This total RNA was then subjected to Nanopore direct RNA sequencing and Illumina total- and polyA-RNA sequencing as described below.

**BKPyV infection and RNA extraction.** Archetype and rearranged BKPyV (Dik and Dunlop, respectively) were purified and titrated as described [47]. RPTE-hTERT cells [48] were plated in 6-well plate and prechilled for 15 min at 4˚C and infected with Dik or Dunlop at a MOI of 1 and 0.5 fluorescence-forming unit (FFU)/cell, respectively. The cells were incubated at 4˚C for 1 h with gentle shaking every 15 min. The virus was removed and fresh REGM medium was added to the cells. Dik and Dunlop infected cells were collected at 120 hpi and 96 hpi, respectively. Total RNA was extracted using the Direct-zol RNA MiniPrep kit (ZYMO Research, USA). This total RNA was then subjected to Nanopore direct RNA sequencing and Illumina total- and polyA-RNA sequencing as described below.

**MPyV infection and RNA extraction.** C57 mouse embryo fibroblasts (ATCC SCRC-1008) were plated on a 150mm dish at 40% confluence. After several hours of growth, the typical DMEM + 10% FBS media was replaced with serum free DMEM. The next day, the crude viral stock was thawed at 37˚C, incubated at 45˚C for 20 minutes to facilitate the final liberation of virus into the supernatant, and cell debris removed from the viral stock with centrifugation. The prepared virus stock was then diluted 1:10 with an absorption buffer consisting of HBSS with 10mM HEPES, 1% FBS, at pH 5.6. Media was removed from the target cells, and 6mL of diluted virus in absorption buffer was added. Infection was allowed to proceed at 37˚C, 5% CO2 for one hour, with the plates rocked every 15 minutes to ensure adequate coverage of the solution over the cell monolayer. At the end of this period, the absorption buffer was removed and DMEM + 2% FBS was added to a final volume of 25mL per 150mm dish. Cells were inoculated for 28 hours at 37˚C, 5% CO2, after which RNA was extracted using TRIzol (ThermoFisher 15596026) according to the manufacturer's instructions. This total RNA was then subjected to Nanopore direct RNA sequencing as described below.

The virus stock used here was kindly provided by the lab of Robert Garcea. This virus stock (viral strain NG59RA) was a crude supernatant from MPyV-infected cells originally generated

by the lab of Thomas Benjamin on 02/08/2011 and was of unknown titer. This stock was subjected to a total of three freeze-thaw cycles before use.

**SV40 polysome profiling.** BSC40 cells were plated on 4 150mm dishes at 60% confluence. After waiting 4 hours for the cells to adhere, cells were infected with SV40 at MOI 1 as reported above. At 44 hours post infection cell culture media was replaced with media containing 100ug/mL cycloheximide and incubated for 5 minutes. Plates were placed on ice, media discarded, and cells were scraped into PBS containing 100ug/mL cycloheximide. Cells were spun down, the PBS discarded, and cells were lysed in a lysis buffer containing 10mM Tris (pH 8), 100mM KCl, 10mM MgCl2, 2mM DTT, 1% Triton X100, 100ug/mL cycloheximide, and 1unit/uL SUPERase RNase inhibitor (Thermo AM2694). Lysates were incubated on ice for 20 minutes with intermittent tapping, and then spun at 10,000g for 10 minutes at 4˚C. The supernatant was loaded onto a 10–55% sucrose gradient followed by ultracentrifugation (Beckman Coulter Optima XPN-100 ultracentrifuge) at 32,500 × rpm at 4˚C for 80 minutes in the SW41 rotor. Gradients were prepared with a gradient mixer and pump. Samples were separated by density gradient fractionation system (Biocomp Piston gradient fractionator IP). RNA was extracted from reserved input ("whole-cell") lysate, as well as the polysome fraction using TRIzol. Equal volumes of each fraction containing heavy polysomes (>2) was pooled prior to extraction [49].

**Western blotting.** Infection of wildtype Dik and two Dik mutants in RPTE-hTERT cells was performed as mentioned above. Protein samples were harvested in E1A buffer with protease and phosphatase inhibitors, electrophoresed, transferred, and probed with large tumor antigen antibody (pAb416) as previously described [48].

**RNA sequencing.** The concentration of total RNA was determined using the Qubit Fluorometer with the Qubit RNA HS Assay Kit (ThermoFisher Q32852). RNA quality was then assessed on an Agilent Bioanalyzer and the RNA 6000 Pico Kit (Agilent 5067–1513). PolyA RNA was isolated using the NEBNext Poly(A) mRNA Magnetic Isolation Module (NEB E7490S) with an input of 5ug of total RNA—for SV40 and BKPyV Dunlop, up to 8 total reactions were used to yield sufficient polyA RNA (500ng) for subsequent protocols. In the case of MPyV, due to limited amounts of total RNA, three reactions were used to yield roughly 100ng of polyA RNA. PolyA RNA concentration was determined again using the Qubit RNA HS Assay Kit (ThermoFisher Q32852). PolyA RNA was then concentrated to 9uL using a centriVap.

500ng of polyA RNA (or, in the case of MPyV, 100ng) in 9uL was then processed using the Nanopore Direct RNA sequencing kit (SQK-RNA002). Resultant libraries were sequenced for up to 24 hours on a MinION using an R9.4.1 flow cell.

In the case of polysome profiling: extracted RNA from the input and polysomes were separately subjected to 5 reactions each of the NEBNext Poly(A) mRNA Magnetic Isolation Module using 5ug RNA input per reaction. All resultant polyA RNA was then processed using the Nanopore Direct RNA sequencing kit (SQK-RNA002). Resultant libraries were sequenced for up to 24 hours on a MinION-Mk1C using an R9.4.1 flow cell.

Illumina total RNA sequencing and polyA RNA sequencing of SV40-, BKPyV Dunlop-, and BKPyV Dik-infected cells was conducted by Novogene Corporation Inc. The QC for the RNA samples was performed using Qubit and Bioanalyzer instruments. Libraries were then prepared using NEBNext Ultra II with RiboZero Plus kit (for short-RNAseq (total)) and NEBNext Ultra II with PolyA Selection kit (for short-RNAseq (polyA)). Both library approaches are strand-specific. Library quality and concentration was assessed with Labchip and qPCR. Libraries were sequenced on NovaSeq6000 using PE150 sequencing.

PacBio SMRT sequencing of SV40 and BKPyV Dunlop was conducted by the Georgia Genomics and Bioinformatics Core. Each sample was subjected to IsoSeq library preparation and sequenced on an individual 8M SMRT cell for 26 hours on a Sequel-II machine.

**Initial sequence processing.** Raw Nanopore dRNAseq reads from standard SV40, BKPyV Dunlop, and MPyV infections were basecalled with Guppy version 4.2.2 with the following command: guppy_basecaller -i fast5 -s basecalled—flowcell FLO-MIN106—kit SQK-RNA002 -r—trim_strategy rna—reverse_sequence true—u_substitution true—cpu_threads_per_caller 10

Raw Nanopore dRNAseq reads from polysome profiling of SV40 transcripts were base-called on a MinION-Mk1C using MinKNOW version 21.02.2.

PacBio SMRTseq subreads were processed using ccs (version 6.0.0). Full-length, nonchimeric reads were then generated using the lima (version 2.0.0) and Isoseq3 (version 3.4.0) packages provided by PacBio.

Stranded Illumina short-RNAseq (total) and short-RNAseq (polyA) reads were processed in the following way: Files containing read 1 (R1) and read 2 (R2) were trimmed and adapters removed using Trim Galore! [50]. Next, reads in R1 files were reverse complemented to orient the reads correctly relative to the transcript of origin, and all read headers in the R1 and R2 files were labeled with "_1" or "_2" respectively. The R1 and R2 files were then concatenated. This Illumina processing pipeline is available in process_illumina.nf.

The MCC tumor RNAseq assessed in this manuscript were short-RNAseq (polyA) that were NOT stranded. This means that the strand of origin of each read is unknown. To address this uncertainty, the complement AND reverse complement of both R1 and R2 were concatenated into the final FASTQ file. As described below in the section "Processing of short-read short-RNAseq (total) and short-RNAseq (polyA) span files", future processing kept the most-likely alignment strand for each read.

**Sequence alignment and processing.** Most long-read sequencing data and Illumina sequencing data were aligned to the appropriate viral genome using Minimap2 [51]. The exceptions are the short-RNAseq (total) JCPyV data from Assetta et al. [40] and the HPyV7 data from Rosenstein et al. [44]—these samples contained sequencing reads of 101bp or shorter and were instead mapped with STAR [52]. All non-primary alignments were discarded. Sequence alignments in BAM format were then converted to BED using bedtools [53]. Here, bedtools considers any Minimap2- or STAR-called intron ("N" cigar flag) as an intron to split alignment segments. Parameters for alignment and bed conversion can be found in minimap2.sh and star.sh.

To capture transcripts that originate from a pre-mRNA that circled the viral genome more than once, and therefore contain repetitive sequences, all alignments were conducted against concatenated copies of the viral genome. In the case of short-read short-RNAseq (total) and short-RNAseq (polyA), the reference consisted of two concatenated copies of the viral genome. For long-read dRNAseq and SMRTseq, the reference consisted of twenty concatenated copies of the viral genome.

Because the references consisted of multiple copies of the same viral genome, mapped reads were assigned to a random copy of the genome. Therefore, all reads in resultant BED files were "slid" such that they started in the first genome copy of the reference using bed_slide_wraparound_reads.py.

All reference genomes can be found in resources/ref directory of the associated github repository. All references used contain the PyV late region at the start/5' end of the reference on the "+" or sense strand, with the early region on the antisense or "-"strand. The concatenated references are based on the following reference genomes collected from NCBI, with any modifications listed:

- SV40: NC_001669.1. The first 100 nucleotides were moved to the end of the sequence.

- BKPyV: KP412983.1

- JCPyV: NC_001699.1

- MPyV: NC_001515.2. The sequence was reverse-complemented to orient the late region towards the start of the reference.

- MCPyV: NC_010277.2

- HPyV7: NC_014407.1

- BSPyV1: LN846618.1

Next, a span file was generated from each slid BED file using bed_to_span.py. This script splits each read into "spans", where each span is an exon or an intron with all positions relative to the viral genome. The introns are defined by the Minimap2- or STAR-called introns ("N" cigar flag) as mentioned above. All regions between the start and end of the reads that are not introns were called as distinct exons. Transcripts were clustered into transcript classes based on introns as discussed below. A "tidy" output span file was then generated that contains the name, strand, and transcript class of a given read, with separate lines for the start and end of each span (e.g., exon or intron) within the sequencing read.

**Alignment of repetitive regions.** Reads that originate from a transcript that circles the genome more than once can be detected because there is one or more repetitive regions within the read. Alignment against multi-copy reference genomes (20 copies in the case of dRNAseq and SMRTseq) as described above sufficiently captured most of these transcripts, with some exceptions. First, BKPyV SMRTseq data had a poor alignment rate of the leader exon in late WA transcripts—this means that WA transcripts are underrepresented in the BKPyV SMRTseq data. Second, alignment of superT and superT* transcripts from SMRTseq and dRNAseq data was generally poor, with the repetitive region often failing to map via Minimap2. Potential superT and superT* reads in dRNAseq and SMRTseq data were identified through assessment of BAM files following mapping. Early reads that contain a CIGAR flag showing an insertion of 100 bases or more were flagged, and up to 50 of these transcripts were manually investigated through online BLASTN [54] against the viral reference genome. Reads supporting superT in SV40 dRNAseq data, superT* in SV40 SMRTseq data, and superT in BKPyV SMRTseq data were initially missing from Minimap2 alignments but were identified via this approach. One transcript of each type was then repaired upon data import to R such that these transcripts are represented in downstream visualizations—these actions are clearly marked in UTILS_import_data.R. Thus, superT and superT* in SMRTseq and dRNAseq data are underrepresented in abundance plots (**S3D and S4D Figs**) and read pileups (**S6 and S8 Figs**) compared to their actual abundance in the cell due to these alignment challenges.

**Generation of transcript classes.** Transcript classes were generated during processing of BED files using bed_to_span.py. Each transcript class consists of sequencing reads that contain the same combination of introns. The transcript class number is based on the abundance of transcripts within a transcript class—e.g., transcript class 1 contains more transcripts than transcript class 2, and so on. Transcript class generation is similar for both long- and short-read sequencing data, although short reads usually (but not always) tend to contain a maximum of one intron. Notably, transcript class assignment is independent of the transcript start and end positions, meaning that there can be heterogeneity of transcript start and end positions within a transcript class. For all SMRTseq and dRNAseq data, for a transcript class to be generated all introns contained within the transcript class were required to be supported by at

least 5 junction-spanning reads within a short-RNAseq (total) dataset. For SV40 and BKPyV Dunlop SMRTseq and dRNAseq data, the short-RNAseq (total) data was generated from RNA from the same extraction. SV40 dRNAseq replicate 2 was corrected with the short-RNAseq (total) data from the first SV40 replicate. For the MPyV dRNAseq data, short-RNAseq (total) data from Garren et al. was used. If a transcript contained an intron that was not supported by at least 5 junction-spanning reads in the Illumina dataset, it was discarded. We opted to use this filtering strategy rather than implementing long-read correction because correction algorithms were unable to cope with wraparound transcripts.

There were limited circumstances where dRNAseq or SMRTseq transcript classes were removed manually during processing—this occurred to four transcript classes that made it through filtering. In these circumstances, alignments were deemed to be artifactual due to Minimap2 alignment errors. These instances are clearly programmatically marked in UTILS_import_data.R with specific rationale for each action.

**Splice coordinate system.** All splice or intron positions marked in any figure or table of this manuscript are **0-indexed positions of the intron**. To convert these coordinates to the 1-indexed/absolute position of the intron on the viral genome, add 1 to the intron start position. For example, for the intron 276–1600, viral genome nucleotide # 277 is the first nucleotide within the intron, and viral genome nucleotide # 1600 is the last nucleotide within the intron.

**Processing of short-RNAseq (total) and short-RNAseq (polyA) span files.** The majority of the short-read RNAseq data investigated here used a strand-specific sequencing strategy (except for the MCC tumor RNAseq). With this strategy, the strand of origin for the transcript yielding each read is known, and a read can be correctly assigned to the sense ("+" / late) or antisense ("-"/ early) strand. However, a fraction of transcripts can be inaccurately stranded due to artifacts during library preparation. When there were many more late reads than early reads in a short-read dataset, a prohibitive fraction of "early" reads would be reads from late transcripts that were incorrectly stranded due to this artifact. To address this issue, short reads that aligned to the + strand were required to either start or end within the late region (defined as the first ½ of the genome), and short-reads that aligned to the—strand were required to either start or end within the early region (defined as the second ½ of the genome).

**Transcript identification.** For **Fig 1** and all supplementary figures, a SV40 or BKPyV transcript was identified and assigned a transcript ID if it was at least 0.1% of early or late strands in dRNAseq or SMRTseq data with one exception—SV40 transcript L8 had been previously identified and was kept despite being at only 0.06% abundance. Existing transcript names, where available, were taken from relevant studies [6,8,14,15]. This assignment occurred from the span files, meaning that all sequencing reads in question were previously required to contain introns that were supported by at least 5 short-RNAseq (total) junction-spanning reads. For SV40, for which there was two dRNAseq replicates, identification of a sequencing read at 0.1% or greater in just one replicate was sufficient.

Transcript IDs (e.g., E1, E2, E3. . ., L1, L2, L3,. . .) consist of the kinetic class (E: Early, or L: Late) of the identified transcript followed by an integer value in ascending order of abundance. This abundance value was calculated by ordering the transcripts in order of the maximum observed relative abundance in dRNAseq or SMRTseq data.

Of note, the relative abundance of transcripts between dRNAseq and SMRTseq data is skewed by distinct read-length biases between the two approaches. The dRNAseq approach has a 3' bias and a bias towards shorter transcripts, while SMRTseq library preparation resulted in preferential sequencing of transcripts closer to ~2500bp in length. Resultant differences in the length of aligned reads can be seen in **S1C Fig**. The TSS distribution of SV40 late

transcripts varies between transcript classes, while the late TSS distribution tends to be similar across transcript classes in BKPyV (**S2A Fig**).

**Calculation of sequencing coverage.** To determine the sequencing coverage for each sample (as in **Fig 1C**), BAM files from alignment were "slid" such that all transcripts must start in the first genome copy of the reference using bam_slide_wraparound_reads.py, in a similar manner as the beds were slid as described above. Forward and reverse strand reads were split, and the depth was calculated using the command 'samtools depth -aa -d0' separately for forward and reverse reads. These processing steps are present in bam_coverage.nf. During plotting, the coverage for each strand was normalized to the maximum coverage at any position (e.g., the maximum coverage of the late and early strands was set to 1).

**Watch plots.** Each panel of a watch plot represents information for a single transcript class. The center "arms" of these plots are histograms detailing the distribution of start (blue) and end (red) positions for the transcripts within the transcript class. These histograms are normalized to the highest abundance position. The outer ring of each watch plot shows the viral ORF map. Each inner grey ring indicates the number of genomes spanned—all transcripts are displayed moving outwards from the center. Red segments indicate the **exons** of each transcript class. The first exon starts on the most-inner grey ring at the most common transcript start site for the transcript class, and the last exon ends on the most-outer grey ring at the most common transcript end site for the transcript class. The 3' end of the transcript is indicated by the red arrow at the end of the last exon. Thus, the transcripts spiral outwards from the center in the direction of the red arrow. **Fig 4** contains a schematic key describing watch plots.

**Read pileup plots.** Each square/rectangular panel of a read pileup plot shows the reads present in a single transcript class. The arrows at the top of each panel indicate the viral ORF map, with dashed lines indicating the end of each genome copy. Next, the lines indicate histograms of the transcript start (teal) and end (yellow) sites for the transcripts within the transcript class. Below the x-axis, each row indicates a single sequencing read. The spans in red indicate the exons inferred from a sequencing read, while the spans in pink indicate the introns/splice junctions. Sometimes the distribution of transcript end positions for a transcript class can be obscured by the thickness of the transcript lines—the histograms should always be consulted to assess abundance.

For SV40 dRNAseq watch and pileups: There were two SV40 dRNAseq replicates. Watch plots and read pileups are based on replicate 1, although missing transcripts that were identified in replicate 2 but not 1 were also plotted.

**Short-read intron plots (S9 and S11 Figs).** In these plots, lines indicate specific introns. The upper and lower horizontal arrows indicate the viral ORF map—often, these ORF maps will indicate two concatenated viral reference genomes. The circles above or below each ORF map indicate the percentage of early or late introns that fall at each genome position. Early introns and percentages are colored red, while late introns and percentages are colored blue.

**polyA tail length.** polyA tail length was determined from dRNAseq data using the 'polya'-command of Nanopolish [55]. To determine the polyA distribution of host transcripts, sequencing reads were aligned to the human GRCh38 (for BKPyV samples), *C. Sabaeus* (for SV40 samples), or mouse (for MPyV) cDNA transcriptomes downloaded from ensembl. Only reads with a Nanopolish QC tag of "PASS" were considered for downstream polyA tail length analyses.

The dRNAseq library preparation included the addition of the "RNA Control Standard" (RCS), which is a synthetic RNA based on yeast ENO2 containing a 30-adenine polyA tail. dRNAseq samples were mapped against ENO2 to assess the polyA tail length distribution of this control.

The cumulative incidence of transcript termination (**S10D Fig**) was calculated by determining, for each early read, how far the read's transcript end site is from the canonical polyA site position for each virus.

**Polysome profiling analysis.**   To determine the ribosome occupancy of host genes, dRNAseq reads were aligned to the *C. Sabaeus* cDNA transcriptome downloaded from ensembl. The number of reads mapped to each transcript was extracted with 'samtools idxstats'. Transcripts were filtered to include only those with at least 10 reads in both polysome and input fractions. The normalized abundance of each transcript in each fraction was defined as (# of mapped reads)/(total number of virus and host mapped reads). Ribosome occupancy of each transcript was determined as (normalized abundance in polysome)/(normalized abundance in whole-cell), where a value of >1 indicates preferential translation.

Ribosome occupancy of individual viral transcripts could not be calculated because of increased rates of transcript truncation in the polysome fraction compared to the whole-cell fraction. This was indicated by a nearly doubled proportion of unspliced reads with premature 5' ends in the polysome fraction compared to the whole-cell fraction, and likely indicates transcript degradation during sucrose centrifugation or fraction collection. Because viral transcripts are mostly identical and vary largely at a 5' splice site, elevated transcript truncation decreased the observed abundance of individual viral transcripts in the polysome fraction and make ribosome occupancy calculations for individual viral transcripts unreliable.

**MCC440 superT PCR and sanger sequencing.**   Anchored poly-dT primers (Life Technologies) were used for specific reverse-transcription of full-length mRNA into cDNA. Primers were designed to uniquely amplify the super-LT junction through exploitation of repetitive sequences. Primer sequences were as follows (5' -> 3'); Forward: CTGGACTGGGAGTCT GAAGC, Reverse: ACCCCTCCTCCATTCTCAAGA. Q5 polymerase (NEB) with standard reaction conditions was used for amplification.

**Generation of integrated PyV structures and viral variant calling.**   Tumor WGS was aligned against a fusion reference genome containing hg38 and Merkel cell polyomavirus (NC_010277) using bowtie2 with default parameters. Integrated virus assembly graphs and annotations were generated using Oncovirus tools (https://github.com/gstarrett/oncovirus_ tools). Assembly graphs were then manually interpreted to create linear integration structures for PyV-associated MCC.

Point mutations were called in the PyV genomes using lofreq with default parameters (https://csb5.github.io/lofreq/) [56]. Lofreq output was functionally annotated with SnpEff (http://pcingola.github.io/SnpEff/) [57] using the relevant GenBank gene annotations for the above genomes. Variants were plotted out in R with the ggplot2 package.

## Supporting information

**S1 Table. Details on SV40 transcripts.**
(XLSX)

**S2 Table. Details on BKPyV Dunlop transcripts.**
(XLSX)

**S1 Fig. Sequencing statistics. A.** The number of reads for all datasets studied here. For long-read dRNAseq and SMRTseq, this number includes spliced and unspliced reads. Because short reads are only useful for transcript characterization when they span a splice junction, the counts for short-reads represent the number of splice-junction-spanning reads. **B.** The cumulative percentage of transcripts in each number of transcript classes, by strand. The X-axis indicated the total number of transcript classes. The Y axis indicates the cumulative percentage of

transcripts within those transcript classes. These plots indicate that most transcripts in most samples are contained within the first few transcript classes. **C-E**. The alignment length distribution of early, late, spliced, and unspliced transcripts for dRNAseq and SMRTseq data from SV40 (**C**), BKPyV Dunlop (**D**), and MPyV (**E**). The X axis indicates the length of a read's alignment, while the Y axis indicates the density/percentage of transcripts with a given alignment length. This plot shows that dRNAseq and SMRTseq data sample from RNA populations of different length.
(TIFF)

**S2 Fig. Transcript start sites and polyA tail lengths. A, B**. The distribution of transcript start sites for late (**A**) and early (**B**) transcripts for SV40 (left column), BKPyV Dunlop (middle column), and MPyV (right column). The arrows indicate the viral ORF positions. **C**. The distribution of polyA tail lengths for the 30-adenine ENO2 control (black), host (red), and viral (yellow) transcripts for SV40, BKPyV Dunlop, and MPyV. The X axis indicates the length of the polyA tail, while the Y axis indicates the density/percentage of transcripts with each length. **D**. Ribosome occupancy of host transcripts in SV40-infected cells. Each grey dot is a host transcript. The red, blue, and black dots are specifically noted host transcripts. Ribosome occupancy is on the Y axis, while the X axis does not hold value. Lines on the violin plot indicate $1^{st}$, $2^{nd}$, and $3^{rd}$ quartiles.
(TIFF)

**S3 Fig. SV40 transcriptome atlas, watch plots. A-C**. Watch plots indicating all identified transcripts in SV40. (**A**) and (**B**) show transcripts that were identified in both dRNAseq and SMRTseq data, while (**C**) shows transcripts identified in SMRTseq only. Pre-mRNA paths are not drawn but can be inferred as indicated in Fig 4. **D**. Barplots that show the abundance of each transcript type in the dRNAseq and SMRTseq data. Here, there are two dRNAseq bars (one per replicate). The Y axis indicates the percentage of transcripts of the same strand. As discussed in the methods, alignment of superT and superT* was challenging, so the actual abundance of these transcripts is higher than reported here.
(TIFF)

**S4 Fig. BKPyV Dunlop transcriptome atlas, watch plots. A-C**. Watch plots indicating all identified transcripts in BKPyV Dunlop. (**A**) and (**B**) show transcripts that were identified in both dRNAseq and SMRTseq data, while (**C**) shows transcripts identified in dRNAseq only. Pre-mRNA paths are not drawn but can be inferred as indicated in Fig 4. **D**. Barplots that show the abundance of each transcript type in the dRNAseq and SMRTseq data. The Y axis indicates the percentage of transcripts of the same strand. As discussed in the methods, alignment of superT and superT* was challenging, so the actual abundance of these transcripts is higher than reported here.
(TIFF)

**S5 Fig. SV40 transcriptome atlas, late transcript read pileups. A, B**. Read pileups showing the late transcripts identified in SV40 dRNAseq (A) and SMRTseq (B). The arrows at the top of the plot indicate the viral ORF positions. Below the X axis, each row is an individual transcript, with exons indicated in red and splice junctions/introns indicated in pink. Above the X axis are histograms indicating the transcript start (teal) and transcript end (yellow) sites. (U: unspliced). Transcript classes are ordered in order of decreasing abundance in each dRNAseq or SMRTseq dataset.
(TIFF)

**S6 Fig. SV40 transcriptome atlas, early transcript read pileups. A, B**. Read pileups showing the early transcripts identified in SV40 dRNAseq (A) and SMRTseq (B). The arrows at the top of the plot indicate the viral ORF positions. Below the X axis, each row is an individual transcript, with exons indicated in red and splice junctions/introns indicated in pink. Above the X axis are histograms indicating the transcript start (teal) and transcript end (yellow) sites. (U: unspliced). Transcript classes are ordered in order of decreasing abundance in each dRNAseq or SMRTseq dataset.
(TIFF)

**S7 Fig. BKPyV Dunlop transcriptome atlas, late transcript read pileups. A, B**. Read pileups showing the late transcripts identified in BKPyV Dunlop dRNAseq (A) and SMRTseq (B). The arrows at the top of the plot indicate the viral ORF positions. Below the X axis, each row is an individual transcript, with exons indicated in red and splice junctions/introns indicated in pink. Above the X axis are histograms indicating the transcript start (teal) and transcript end (yellow) sites. (U: unspliced). Transcript classes are ordered in order of decreasing abundance in each dRNAseq or SMRTseq dataset.
(TIFF)

**S8 Fig. BKPyV Dunlop transcriptome atlas, early transcript read pileups. A, B**. Read pileups showing the early transcripts identified in BKPyV Dunlop dRNAseq (A) and SMRTseq (B). The arrows at the top of the plot indicate the viral ORF positions. Below the X axis, each row is an individual transcript, with exons indicated in red and splice junctions/introns indicated in pink. Above the X axis are histograms indicating the transcript start (teal) and transcript end (yellow) sites. (U: unspliced). Transcript classes are ordered in order of decreasing abundance in each dRNAseq or SMRTseq dataset.
(TIFF)

**S9 Fig. Intron plots for all datasets studied. A.** Intron plots generated from short-read RNAseq. The arrows at the top and bottom of each panel indicate the position of viral ORFs. The lines indicate specific introns identified in the RNAseq data, with the 5' end on the top and the 3' end on the bottom. The blue color indicates late transcripts, with red indicating early transcripts. The size of the circles above and below the viral ORF maps indicate the percentage of junction-spanning reads with a 5' end (on top) or 3' end (on bottom) at that position. Junctions are plotted if they are at least 1% of early or late transcripts, except for the SV40 pA superT junction (transcript class 3) which is just below threshold but is of interest. **B.** Another representation of intron plots for each virus. The top arrows indicate the position of viral ORFs. The X axis indicates the genomic position for each splice. The Y axis indicates a single transcript class, with that class' intron plotted as a line. The percentage of early or late transcripts is indicated with the numeric percentage. Junctions are plotted if they are at least 1% of early or late transcripts, except for the SV40 pA superT junction (transcript class 3) which is just below threshold but is of interest. Transcript classes with labels surrounded by a red box indicate that they lack either the 5' GT (donor) or 3' AG (acceptor) splice site dinucleotides.
(TIFF)

**S10 Fig. Alternative polyadenylation of early transcripts in SV40, BKPyV, and MPyV. A-C**. Watch plots indicating the LT and ST transcripts for SV40 (**A**), BKPyV Dunlop (**B**), and MPyV (**C**). The focus of these plots is the distribution of transcript end positions, which are the inner red arms. The region of APA of highlighted in blue, with the canonical transcript end sites highlighted in red. **D**. A cumulative incidence plot of transcript termination in SV40 (blue), BKPyV Dunlop (red), and MPyV (green). The X axis indicates the distance to the

canonical polyA site, while the Y axis indicates the percentage of transcripts that have terminated by that position. **E-G**. Density plots showing the distribution of polyA tail lengths for LT and ST transcripts that end at the canonical site (solid) or undergo APA (dashed) for SV40 (**E**), BKPyV Dunlop (**F**), and MPyV (**G**). The x axis indicates the length of the polyA tail, while the Y axis indicates the density/proportion of transcripts with the given length. **H-I**. Watch plots indicating the LT and ST transcripts from polysome-associated (**H**) or whole-cell (**I**) RNAs. The focus of these plots is the distribution of transcript end positions, which are the inner red arms. The region of APA of highlighted in blue, with the canonical transcript end sites highlighted in red.
(TIFF)

**S11 Fig. short-RNAseq (polyA) analysis of BKPyV Dik WT, M1, and M2. A-C**. Intron plots generated from short-read (polyA) RNAseq of cells infected with BKPyV WT, or the M1 or M2 mutants. The arrows at the top and bottom of each panel indicate the position of viral ORFs relative to the standard BKPyV genome—note that the genomes of mutants M1 and M2 are altered as indicated in Fig 5E. The lines indicate specific introns identified in the RNAseq data, with the 5' end on the top and the 3' end on the bottom. The size of the circles above and below the viral ORF maps indicate the percentage of junction-spanning reads with a 5' end (on top) or 3' end (on bottom) at that position. Only early junctions that are at least 1% of early transcripts are plotted. The superT junction is colored in gold. (**A**) Intron plot for BKPyV Dik WT. (**B**) Intron plot for BKPyV Dik M1. (**C**) Intron plot for BKPyV Dik M2.
(TIFF)

**S12 Fig. superT in MCPyV-associated MCC. A.** Sanger sequencing of an RT-PCR product from MCC J45_440, showing the superT-specific junction. **B.** A schematic detailing the MCC 285 MCPyV integration site, showing how it is possible that superT is generated via cis-splicing. **C.** The assembled viral block in MCC tumor J45_440. This integration site is based on de-novo assembly using short whole genome sequencing reads. Despite only assembling one viral block, we found that 1) there are likely 2 copies of the viral genome, and 2) the 5' viral integration site appears to fall on chromosome 7 "after" the 3' viral integration site, observations consistent with the existence of two copies of the viral genome in tandem separated by a small segment of host DNA at this integration site. **D.** The assembled viral blocks in MCC tumor J17_296. The longest block contains two copies of the early region. **E.** Lollipop plots showing identified SNPs in the MCPyV genomes of J45_440, J17_296, and J11_285. The gene-map below the Fig indicates the position of viral ORFs. Each lollipop is colored according to the nucleotide substitution identified.
(TIFF)

## Acknowledgments

We thank Katie Vicari (www.KatieRisVicari.com) for her work illustrating Figs 1A, 2, 3, and S12B. We thank Mary O'Reilly with the Broad Research Communication Lab for helpful discussions on figure design. We thank Robert Garcea and Kimberly Erickson for kindly providing stocks of MPyV. Portions of this research were conducted using the O2 High Performance Computing Cluster, supported by the Research Computing Group at Harvard Medical School. Portions of this work utilized the computational resources of the NIH HPC Biowulf cluster (http://hpc.nih.gov).

## Author Contributions

**Conceptualization:** Jason Nomburg, Matthew Meyerson, James A. DeCaprio.

**Data curation:** Jason Nomburg.

**Formal analysis:** Jason Nomburg, Gabriel J. Starrett.

**Funding acquisition:** Shobha Vasudevan, Michael J. Imperiale, Matthew Meyerson, James A. DeCaprio.

**Investigation:** Jason Nomburg, Wei Zou, Thomas C. Frost, Chandreyee Datta, Gabriel J. Starrett.

**Methodology:** Jason Nomburg, Matthew Meyerson, James A. DeCaprio.

**Project administration:** Jason Nomburg.

**Resources:** Jason Nomburg, Wei Zou, Thomas C. Frost, Chandreyee Datta, Shobha Vasudevan, Gabriel J. Starrett, Michael J. Imperiale, Matthew Meyerson, James A. DeCaprio.

**Software:** Jason Nomburg.

**Supervision:** Matthew Meyerson, James A. DeCaprio.

**Validation:** Jason Nomburg.

**Visualization:** Jason Nomburg.

**Writing – original draft:** Jason Nomburg, Matthew Meyerson, James A. DeCaprio.

**Writing – review & editing:** Jason Nomburg, Wei Zou, Thomas C. Frost, Chandreyee Datta, Shobha Vasudevan, Gabriel J. Starrett, Michael J. Imperiale, Matthew Meyerson, James A. DeCaprio.

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
