## [Decision Letter · Decision Letter 0]

7 Feb 2022

Dear Jim,

Thank you very much for submitting your manuscript "Long-Read Sequencing Reveals Complex Patterns of Wraparound Transcription in Polyomaviruses" for consideration at PLOS Pathogens. As with all papers reviewed by the journal, your manuscript was reviewed by members of the editorial board and by several independent reviewers. The reviewers appreciated the attention to an important topic. Based on the reviews, we are likely to accept this manuscript for publication, providing that you modify the manuscript according to the review recommendations.

Sincerely,

Walter J. Atwood

Associate Editor

PLOS Pathogens

Karl Münger

Section Editor

PLOS Pathogens

Kasturi Haldar

Editor-in-Chief

PLOS Pathogens

orcid.org/0000-0001-5065-158X

Michael Malim

Editor-in-Chief

PLOS Pathogens

orcid.org/0000-0002-7699-2064

Reviewer Comments (if any, and for reference):

Reviewer's Responses to Questions

**Part I - Summary**

Reviewer #1: Nomburg et al have provided a comprehensive analysis of transcription patterns in diverse polyomaviruses. The care and detail applied to this work represents a real tour de force and offers a range of exciting insights into the biology of these underappreciated pathogens. While the general transcriptional analysis is compelling, I was particularly intrigued by the analysis of SuperT, which really served to highlight the power of the methodology. Overall, the manuscript is well written and the reams of supporting data are readily accessible. As such, I only have a few significant critiques/comments, which I think would improve the manuscript, as well as a number of minor comments. Assuming these are addressed then I would highly recommend this manuscript for acceptance.

Reviewer #2: This is an excellent transcriptomic study performed Nomburg et al. on multiple diverse polyomaviruses. The authors use a combination of long and short format high throughput sequencing of polyA-enriched RNAs. The combination of techniques combined with the clever application of evolutionary conservation makes for one of the most diverse comprehensive evolutionary comparative study of viral transcriptomics I am aware of. The authors use both of these traits (detection in alternative methods and/or evolutionary conservation) to identify new transcripts. They then confirm that at least some of these newly identified transcripts are likely translated since they are found in polysomal-enriched fractions of RNA. Although no function is provided for the newly identified transcripts, the fact that some are found on polysomes serve as useful hypothesis generators worthy of future study. Overall, this work will likely be a useful study to many polyoma virologists for years to come.

Major comment:

I could not find reference to important Acheson paper that first described the phenomena of "giant RNAs", or wrap around transcripts further elucidated here. Minimally, this needs to be cited (I apologize if I missed it somehow), and I suggest that it warrants a sentence or two in the Intro and Discussion: Acheson NH. Polyoma virus giant RNAs contain tandem repeats of the nucleotide sequence of the entire viral genome. Proc Natl Acad Sci U S A. 1978 Oct;75(10):4754–4758.

Minor comment:

1. For the JC MT-like transcripts, are any of these predicted to contain transforming motifs of MuPyV MT?

**Part II – Major Issues: Key Experiments Required for Acceptance**

Reviewer #1: Reviewers note: There are no additional key experiments required from my perspective. However, I do think it is imperative that the following analyses are performed or, if not possible, appropriately caveated in the text. It is also imperative to ensure all sequencing data (inc. raw fast5 files) are made available at the time of publication.

• From the bioproject page, it appears only the FASTQ datasets derived from Nanopore DRS have been uploaded. Given the rapid improvements in basecalling associated with nanopore, the authors should really upload the raw fast5 data. This can be done via the European Nucleotide Archive in a relatively simple manner (make a tarball of the raw fast5 data and upload) and should be included under the bioproject ID.

• One element that I think is lacking from the results is a clear analysis of how abundant each transcript is. While I kind roughly estimate this for SV40 from Figure 1G, it would be very useful to have this information for all the viruses. Unsurprisingly there is a large dynamic range and it looks like novel transcripts are generally not very abundant but this is not always clear.

• Some further analysis of the polysome sequencing datasets is warranted. It would be particularly interesting to know whether transcripts originating from wraparound transcription are also translated and if so, how their loading compares to raw abundance.

• How many of the described features (e.g. APA, wraparound transcription) are temporally regulated? Do these only arise during Late stages or would they also be expected at early timepoints? i.e. can the authors justify not profiling multiple distinct timepoints during infection.

Reviewer #2: (No Response)

**Part III – Minor Issues: Editorial and Data Presentation Modifications**

Reviewer #1: • Figures 2 & 3 are beautifully constructed but I could not adequately distinguish between Acceptor (pink?) and Exon (red). I would suggest changing one of those colors to better distinguish these features.

• Lines 57-63 in the introduction would benefit from a few citations.

• Line 130 – I think this should refer to Figure S2?

• Figure S9 would benefit from an analysis of splice donor and acceptor motifs. It would be useful to then correlate rare/non-canonical motifs against transcript abundance.

• ‘In contrast’ is used frequently but is not always grammatically correct. Consider also using ‘By contrast’ where appropriate.

• It’s not clear to me why the authors utilized the NEB poly(A) selection approach given it’s limited loading capacity? Using a larger scale poly(A) purification approach (e.g. Invitrogen Dynabeads) would obviate the need for vacuum-based concentration.

• Can the authors confirm that all transcript isoforms observed contained measurable poly(A) tails? It is often beneficial to omit DRS reads without defined poly(A) tails (as reported by nanopolish) to reduce potential artefacts.

• It would help to order the transcript boxes shown in Fig S5A-B so that they occupy the same relative space (e.g. lower left should be L11_Novel in both A&B) as this makes visual comparison between DRS and PacBio results much easier. I’m also not sure what to make of the ‘read-through’ reads observed in several of the PacBio samples (e.g. U, L1_G)? Is this an artefact? I also find the use of red (TSS) and blue (PAS) to be slightly confusing in the context of also showing ORFs above in red and blue.

• It might be a simple formatting issues but I do not see any of the pre-mRNA paths shown on any of the watch plots except for the legend schematic in Fig4). While this might also be intentional (i.e. I can roughly infer this for each plot), it would be useful to have those pre-mRNA paths shown on each individual plot.

Reviewer #2: (No Response)

PLOS authors have the option to publish the peer review history of their article (what does this mean?). If published, this will include your full peer review and any attached files.

Reviewer #1: No

Reviewer #2: No

Figure Files:

Data Requirements:

Reproducibility:

References:

---

## [Editor Report · Decision Letter 1]

27 Feb 2022

Dear Dr. DeCaprio,

We are pleased to inform you that your manuscript 'Long-Read Sequencing Reveals Complex Patterns of Wraparound Transcription in Polyomaviruses' has been provisionally accepted for publication in PLOS Pathogens.

Best regards,

Walter J. Atwood

Associate Editor

PLOS Pathogens

Karl Münger

Section Editor

PLOS Pathogens

Kasturi Haldar

Editor-in-Chief

PLOS Pathogens

orcid.org/0000-0001-5065-158X

Michael Malim

Editor-in-Chief

PLOS Pathogens

orcid.org/0000-0002-7699-2064
---

## [Editor Report · Acceptance letter]

29 Mar 2022

Dear Dr. DeCaprio,

We are delighted to inform you that your manuscript, "Long-Read Sequencing Reveals Complex Patterns of Wraparound Transcription in Polyomaviruses," has been formally accepted for publication in PLOS Pathogens.

Best regards,

Kasturi Haldar

Editor-in-Chief

PLOS Pathogens

orcid.org/0000-0001-5065-158X

Michael Malim

Editor-in-Chief

PLOS Pathogens

orcid.org/0000-0002-7699-2064